# Validity of a Magnet-Based Timing System Using the Magnetometer Built into an IMU

**DOI:** 10.3390/s21175773

**Published:** 2021-08-27

**Authors:** Carla Pérez-Chirinos Buxadé, Bruno Fernández-Valdés, Mónica Morral-Yepes, Sílvia Tuyà Viñas, Josep Maria Padullés Riu, Gerard Moras Feliu

**Affiliations:** 1National Institute of Physical Education of Catalonia (INEFC), University of Barcelona (UB), 08038 Barcelona, Spain; cperezchirinos@gencat.cat (C.P.-C.B.); bfernandez-valdes@tecnocampus.cat (B.F.-V.); monicamorralyepes@gmail.com (M.M.-Y.); siltuvi22@gmail.com (S.T.V.); jmpadu@gmail.com (J.M.P.R.); 2School of Health Sciences, TecnoCampus, Pompeu Fabra University, 08302 Barcelona, Spain

**Keywords:** inertial measurement unit, wearable sensor, timing system, gate crossing time, performance, sports biomechanics, running, skiing

## Abstract

Inertial measurement units (IMUs) represent a technology that is booming in sports right now. The aim of this study was to evaluate the validity of a new application on the use of these wearable sensors, specifically to evaluate a magnet-based timing system (M-BTS) for timing short-duration sports actions using the magnetometer built into an IMU in different sporting contexts. Forty-eight athletes (22.7 ± 3.3 years, 72.2 ± 10.3 kg, 176.9 ± 8.5 cm) and eight skiers (17.4 ± 0.8 years, 176.4 ± 4.9 cm, 67.7 ± 2.0 kg) performed a 60-m linear sprint running test and a ski slalom, respectively. The M-BTS consisted of placing several magnets along the course in both contexts. The magnetometer built into the IMU detected the peak-shaped magnetic field when passing near the magnets at a certain speed. The time between peaks was calculated. The system was validated with photocells. The 95% error intervals for the total times were less than 0.077 s for the running test and 0.050 s for the ski slalom. With the M-BTS, future studies could select and cut the signals belonging to the other sensors that are integrated in the IMU, such as the accelerometer and the gyroscope.

## 1. Introduction

Time is one of the decisive parameters of performance in many sports [1,2]. In individual sports, such as athletics (running) or winter sports disciplines such as alpine skiing, time will determine the position in the ranking. In alpine skiing, for example, the ranking difference depends on hundredths of a second [3,4], and any small detail can change the standings. In 2020, for instance, at the FIS Alpine Ski World Cup in Madonna di Campiglio (Italy), the difference between first and fourth place in the overall time was less than half a second. Moreover, among the top ten skiers, there were differences of less than 1 s [5].

Differences of hundredths of a second in a one-minute race, following the example above, represent a percentage of less than 0.03% of the total time. Therefore, knowing the time elapsed between the start and the finish line does not provide sufficient information with which to qualitatively evaluate a skier’s performance along the course [6,7]. In athletics, the same occurs, the total time is too general, and therefore it is common to train with intermediate times [8]. For example, a 100-m race is divided in different sections and the time of each section is obtained. In this way, the coach has information on each phase of the race: acceleration, maximal velocity and reduction of velocity [9,10]. In skiing, intermediate times are often measured during training and competition contexts, usually a middle time and a final time to delimit changes of slope or changes of pace within the course [11]. However, it has been shown that these intermediate times still provide too general information to explain performance in the case of skiing [6]. It has been noted that gate-to-gate times can provide additional data to further analyze skiers’ performance [12,13]. Although it is not possible to officially obtain more information on gate crossing times during competitions, it is possible to obtain this information during training sessions.

In technical skiing disciplines such as slalom and giant slalom, the time between gates can range from less than one second to less than two seconds [14], the same time it can take to complete a 10-m race [8]. Therefore, to measure this type of short-duration actions accurate and reliable measurement systems would be needed. Currently, one of the most widely used timing technologies in these sport contexts is the photocell [15]. Electronic timing provides an accurate and instantaneous result, which makes it easy to quickly compare a large group of skiers or athletes. Because of these characteristics, this is the timing equipment used in official competitions governed by FIS regulations [16] and it is also one of the reference systems used as gold standard in sport research [12,13,17,18,19]. However, some of the limitations include: the time taken for correct positioning and alignment, the high economic cost, the weather conditions that can affect time measurements, e.g., extreme temperatures [16], the fact that each athlete can cut the light beam with a different part of the body [20], and the reduced number of times that could be taken along a course in relation to the number of gates, as in the case of skiing. Another system used for accurate time measurement is recording with video or high-speed cameras, measuring between 100 and 1000 Hz [15]. Video and photo finish are the official timing systems in high-level athletics competitions [15,21]. Despite the high precision that can be obtained, it is a system that requires considerable time to set up since it is necessary to provide and align visual references in order to correct the error that the observer may have [22]. In the case of skiing, the use of video analysis could limit the number of gates to be analyzed [2,23,24]. In addition, the fact that time is not presented immediately, since a software is needed to analyze the recordings makes it not a field method for training days and its use is more suitable for research [15].

In recent years, there have been authors who have validated other methods to control time in a more practical, economic, and detailed way. Supej et al. [12] validated a new method for time computation from surveyed trajectories using a high-end global navigation satellite system (GNSS). They validated it in two different contexts: running and skiing. This system, however, involves carrying a backpack on the back with a light weight antenna that could influence the skier’s technique [25]. In addition, it can only be used outdoors, and an alternative must be found for indoors. Fasel et al. [13] validated a magnet-based timing system (M-BTS) for detecting gate-to-gate time in alpine ski racing using the magnetometer of an inertial measurement unit (IMU) and several magnets. This is a lighter, more ecological, and easier to use alternative method for the measurement of multiple times over long distances in alpine skiing and other sports, regardless of the place where they are practiced. In addition, the use of IMUs in sports training has made it easier to obtain kinematic and kinetic data on movement [26,27,28,29,30,31]. The fact that all the built-in sensors (accelerometer, gyroscope, GNSS receiver, magnetometer, etc.) are synchronized optimizes data collection. However, nowadays there are IMUs from different manufacturers with different characteristics such as the sampling frequency used, which can affect their validity.

To our knowledge, no study has explored the validity of a M-BTS using the magnetometer built into an IMU in different sporting contexts. Therefore, the aim of this study was to evaluate the validity of a M-BTS for timing short-duration sport actions using a an IMU’s magnetometer in a linear sprint running test and in an alpine ski slalom.

## 2. Materials and Methods

### 2.1. General Overview

Two independent experiments were carried out. The first involved athletes performing a linear sprint running test in a soccer field, to assess the validity of the proposed system in an athletic environment. The second involved skiers performing a slalom on a ski slope. The two experiments pursued the same aim but under different conditions (Figure 1).

### 2.2. Subjects

Athletes: Forty-eight experienced, recreational level athletes (22.7 ± 3.3 years, 72.2 ± 10.3 kg, 176.9 ± 8.5 cm, 109.9 ± 7.7 cm trochanteric height, 7.2 weekly training hours) participated in the study.

Skiers: Eight alpine skiers (17.4 ± 0.8 years, 176.4 ± 4.9 cm, 67.7 ± 2.0 kg, 128.8 ± 26.6 slalom FIS-Points) participated in the study.

Written informed consent was obtained from all participants or by their parents in case they were under 18 years of age. All the procedures were approved by the Ethics Committee for Clinical Sport Research of Catalonia (Study Number: 27/CEICGC/2020) and were conducted in accordance with the Declaration of Helsinki.

### 2.3. Procedures

In both experiments, an IMU device (WIMU, Realtrack Systems, Almeria, Spain) weighing 70 g and with a size of 81 mm × 45 mm × 15 mm was attached to the lower back of athletes and skiers, at the L4–L5 level, using an adjustable sports lycra belt (Figure 2).

This location close to the center of gravity (CG) can be easily attached and detached and it is a comfortable place to carry by athletes. Besides, it has been shown that this is the best place to assess whole body movement and to detect ski turns [32,33]. The wearable sensor also contains a 3D magnetometer recording at 100 Hz which allows a resolution of hundredths of a second. It uses anisotropic magnetoresistive technology. The units of measurement are milligauss, and its maximum scale is ±8 Gauss. When the IMU is turned on it chooses the scale that creates the best for the environment it is located in at that moment, the typical is between ±2 Gauss and ±4 Gauss. For the two experiments, IMU calibration was performed on a flat and even surface with the *z*-axis perpendicular to the surface, according to the manufacturer’s specifications.

Bar magnets of diameter 33 mm and height 267 mm (D33 mm × 267 mm, ND35, A.C. magnets 98, Barcelona, Spain) were designed and used for both experiments.

A standard time keeping system based on photocells (Witty System, Microgate, Italy) was used to validate the M-BTS. A reflector-type single-beamed photocell system was used, where the photocell had the transmitter and receiver electronics in the same case. A simple reflector on the opposite side was used to reflect the photocell beam back to the main unit. This placement of the photocells is reminiscent of a gate that the athlete/skier will have to cross. Hence the section time is also called gate crossing time. A resolution of thousandths of a second was obtained.

In both experiments, several section times were obtained instantly with the photocells and noted. The section times collected by the M-BTS were not obtained instantly but were calculated afterwards. Each time an athlete or skier passed near a bar magnet, a peak appeared in the magnetometer time series. SPRO software (Realtrack Systems, Almeria, Spain) was used to download the data stored in the IMU in order to calculate the elapsed time between peaks (Figure 1). The magnetometer source signal was used as the dominant peaks corresponded to the gate locations. Since the magnets were placed vertically, the magnetic field came from the ground upwards. For this magnet placement, the vertical *x*-axis of the IMU magnetometer was the one that collected the highest magnitude peaks and was therefore selected. The signal was loaded into the program workspace and the selection tool was used to select the signal chunks corresponding to the sprints and the downhill ski slaloms to be analyzed. The SPRO peak detection tool was then manually configured to detect the maximum positive peaks lasting between 200 and 2000 ms and the time between peaks was automatically detected.

#### 2.3.1. Experiment 1

Athletes were asked to complete a 60-m linear sprint running test at maximum speed. All test sessions were performed at an outdoor grass soccer field. Prior to the start, participants were instructed to perform a 15 min warm-up of their choosing and then performed the test. The 60-m linear sprint running test was divided into five different sections: the 0–5 m section, the 5–10 m section, the 10–20 m section, the 20–40 m section, and the 40–60 m section. Six bar magnets were used to cover all the distance. Bar magnets were placed at the beginning and at the end of each section, as detailed in (Figure 1a). After observing that there were no changes in the magnetic field, bar magnets were placed on top of a metal base of a 5 kg weight disk to ensure their fastening (Figure 1a). They were positioned vertically with the north pole pointing upwards. Previous tests in the laboratory revealed that the bar magnets distort the magnetic field up to a distance of about 1 m. Therefore, in order to ensure that the athletes passed close enough, a corridor of no more than 1 m wide was chosen. Six sets of photocells were placed in line with the magnets and were set approximately at hip height according to the recommendations of Yeadon, Kato, and Kerwin in 1999 for a single beam system [20]. 

This setup allowed to obtain five intermediate times-corresponding to each section-for each system: photocells and M-BTS. Referring to this last one, six peaks were detected for each trial in the magnetometer signal. The total time was defined as the time sum of the five sections.

Five athletes were discarded due to poor detection of the first peak in the magnetometer signal. This occurred because the starting line of the running test coincided with the position of the first bar magnet. After these subjects, the starting line was modified and moved 2 m forward from the first bar magnet (Figure 1a).

During the four days of data collection, an average air temperature of 10 °C was recorded, with minimum and maximum temperatures of 7 °C and 13 °C, respectively. The maximum wind speed recorded was 0.02 km/h and was not considered.

#### 2.3.2. Experiment 2

Skiers were asked to complete a run in a 40-gate slalom course at maximum speed. Only gates 21–24, hereinafter referred to as gates 1–4, were considered for the study (gate distance: 10.5 m; gate offset: 3.5 m). All gate distances and course settings were according to the International Ski Federation (FIS) rules [34]. The four gates formed 3 sections: the first section between gates 1 and 2, the second section between gates 2 and 3, and the third section between gates 3 and 4. Following the guidelines set by Fasel et al. [13], on the inner side of each gate’s turning pole and very close to the base, four bar magnets were buried vertically into the snow such that the top was slightly below the snow surface (Figure 1b). Four sets of photocells were used to measure the reference time and were installed 10 cm above gates 1–4 and below knee height, in accordance with the FIS rules [34]. In order to avoid any unnecessary risk of injury, a 7 m width between the transmitter/receiver and the reflector was set.

This setup allowed to obtain three intermediate times, corresponding to each section, for each system: photocells and M-BTS. Referring to this last one, four peaks were detected for each run in the magnetometer signal. The total time was defined as the time sum of the three sections.

During data collection, the snow and air temperatures progressively increased from −7.1 to −2.4 °C and from −4.1 to −2.3 °C, respectively. The maximum wind speed recorded was 5.2 km/h in a south-westerly direction, thus perpendicular to the direction of the ski run.

### 2.4. Statistical Analzysis

In both contexts, sample distributions were tested for normality with a Shapiro–Wilk test. In the running, a dataset of 43 trials, one trial for each athlete, was used for validation. The same criteria were followed for skiing, resulting in a dataset of 8 runs.

The criterion validity of time measured by the M-BTS was assessed using mean time differences and the 95% error-range against actual time obtained from photocells. Mean time differences were defined as the reference system’s values minus the proposed system’s values, and the 95% error-range was defined as the range between the 2.5th and 97.5th percentiles. Bland–Altman plots were used to graphically complement the differences between the two systems [35]. Additionally, the intraclass correlation coefficient (ICC) with corresponding 95% confidence intervals (CI) and linear regression analysis were used for agreement evaluation between both systems.

Data analysis was conducted using PASW Statistics 21 (SPSS, Inc., Chicago, IL, USA). It was set as excellent agreement that the ICC using a two-way mixed-effects model was ≥0.75 [36]. The significance level was set at 0.05 for all analyses.

## 3. Results

Data did follow normal distribution. Section times and total times provided by photocells and the M-BTS and the mean differences between both systems, are presented in Table 1. Differences between the two measurement systems were no significant in any of the contexts, neither in overall section times (Running *p* = 0.559, Skiing *p* = 0.880), nor in the total time (Running *p* = 0.684, Skiing *p* = 0.884). All data can be accessed in the Appendix A.

In the running context, the overall section time 95% error-range was (−0.024 s; 0.068 s). The 95% error-range for the total time was (0.011 s; 0.087 s). In the case of skiing, the overall section time’s 95% error-range was (−0.079 s; 0.061 s). The 95% error-range for the total time was (−0.035 s; 0.015 s) (Table 1).

The Bland–Altman plots visually showed the differences between the two systems for the different section times and for the total time in both sport contexts (Figure 3). In the context of running, a lower bias (0.011 ± 0.021 s) and a narrower LOA (−0.030–0.052 s) were observed when analyzing all sections compared to total time (Bias: 0.054 ± 0.023 s; LOA: 0.008–0.099 s). For skiing, the total time obtained a slightly higher bias (−0.013 ± 0.020 s) and a narrower LOA (−0.052–0.026 s) than the analysis of all sections (Bias: −0.004 ± 0.043 s; LOA: −0.089–0.080 s). Linear regression analysis confirmed these results with high levels of model explanation for the two sport contexts.

Excellent ICCs between measured and actual time were found in all sections (Running: 1.00, 95% CI = 0.99–1.00; Skiing: 0.95, 95% CI = 0.88–0.98) and in the total time (Running: 1.00, 95% CI = 0.99–1.00; Skiing: 0.99, 95% CI = 0.98–0.99).

## 4. Discussion

To our knowledge, this is the first study evaluating the validity of a M-BTS using the magnetometer built into an IMU in different sporting contexts. Two independent experiments were carried. The first took place in a soccer field with athletes performing a linear sprint running test and the second was conducted on a ski slope with skiers performing a slalom. The results demonstrate that M-BTS is valid for measuring short-duration sport actions about one second and above in different sport contexts.

In this study, the mean difference in all sections between the two systems obtained was 0.011 ± 0.021 s for the linear running test and −0.004 ± 0.043 s in the ski slalom (Figure 3). These results have increased the mean difference found in Fasel et al. (Skiing: −0.002 ± 0.005 s) [13] and Supej et al. (Running: 0.0005 ± 0.0070 s; Skiing: 0.0002 ± 0.0001 s) [12]. In relation to the 95% error-range for the total time it was 0.077 s for running and 0.050 s for skiing, which have also increased when compared with the 95% error-range obtained by the studies mentioned above [12,13]. These differences could be explained by several factors. Firstly, the sampling frequency of the IMU’s magnetometer, which recorded at 100 Hz, provided a resolution of 0.01 s. For the highest velocity recorded in this study (up to 38 km/h), the resolution of the IMU’s magnetometer resulted in an unmeasured displacement of approximately 10.5 cm. In slalom, maximum speeds of up to 55 km/h are usually reached [14], which would represent an unmeasured displacement of approximately 15 cm. A possible solution would be to increase the sampling frequency by linear interpolation if that degree of accuracy is needed [13]. Along these lines, the GNSS timing system developed by Supej et al. already used interpolation to detect the exact passage through the gate [12,37].

The second factor could be the speed. This parameter is a double-edged sword, since in the M-BTS, the peaks recorded at higher speed reduce their width, which allows for greater exactitude in their location [13]. The influence of the speed factor could be detected in the case of the linear running test in which each section had a different speed and a tendency for the dispersion of the time differences to decrease with increasing speed was found (Figure 3). In the case of skiing, the sections were very similar to each other in terms of distance and speed. Section 2 was slightly faster but no differences due to speed were observed in scatter plots. However, lower speeds (up to about 38 km/h) have been recorded compared to those obtained in the study conducted by Fasel et al. (above 50 km/h) [13], which could explain why they found the lowest mean differences between the M-BTS and the photocells. 

Remaining in the context of skiing, another factor that could have increased the differences between the results obtained in the present study compared to Fasel et al. [13] is the level of the skiers. A higher level will determine the correct technical execution and approach to the slalom pole and consequently to the bar magnet. As mentioned above, the distance from the bar magnet is an important factor, the closer the magnet is passed, the greater the magnetic field distortion recorded by the IMU’s magnetometer. In this sense, it could be deduced that the higher the level, the better the proposed timing system will work.

Therefore, since the times obtained through the proposed M-BTS have relatively small mean differences in section and total times and a narrow dispersion compared to the photocell measurements, the proposed system could be used instead of the photocells during regular trainings in different sport contexts considering that subtle performance changes below reported 95% error-ranges would not be detected.

Advantages of M-BTS: Compared to photocells, the M-BTS involves fixing the IMU in a body location, preferably the same for all subjects, and this eliminates the problem of athletes randomly breaking the photocell beams with different parts of the body [20]. The time triggering always occurred in the same situation: when the IMU passes closest to the magnet. As a curiosity, in the linear running test, in 70% of the cases M-BTS underestimate the time measured by photocells, and in 30% of the cases photocells have reported a shorter time than the M-BTS. These could be cases in which the beam could be triggered early by swing arms. Yeadon et al. [20] described a similar percentage in their study with photocells positioned at hip height. In the case of skiing, however, the M-BTS have overestimated and underestimated the photocell values, by 60% and 40%, respectively. Placing the photocells lower and 10 cm forward with respect to the turning pole could have reduced false time-triggering with other body parts or with the pole itself.

The use of magnets instead of photocells reduces preparation time for coaches. No time is lost in the correct alignment of the transmitter/receiver with the reflector. This alignment is very important as it should be perpendicular to the direction of movement to minimize errors [12]. In linear movements, such as a sprint, this alignment is easier. However, in curvilinear movements it is more complex to determine the perpendicular orientation of the movement since for each run and skier will be different. 

In addition, the M-BTS reduces the risk of an accident due to a collision with the photocells. Especially in the context of skiing where there is an official regulation on how to build and place the wooden posts that will hold the photocells [16]. This could also be of interest for team sports or sports involving implements such as balls, where ecological systems that modify the environment as little as possible are needed. Moreover, the M-BTS allows for more intermediate times, such as gate-to-gate times, which will give detailed performance information [12,13,37].

Compared to the GNSS-based approach, the proposed system configuration is more economical and easier to use for trainings, since no differential GNSS with a base station is needed and the position of each gate is not required to be surveyed. Except for the magnets, no additional hardware is requested. It is also a more ecological system for athletes/skiers than having to carry a backpack, as it only requires wearing a small lycra belt to put on the wearable device. In addition, it can be used anywhere, indoors or outdoors, and is not affected by adverse weather conditions.

Finally, the fact of using the M-BTS will allow to select the peaks obtained from the magnetometer signal as a marker to cut the signal of any of the other sensors such as the accelerometer, the gyroscope and the GNSS receiver, all of them integrated in the IMU. Often, it is difficult to find where a sports action begins and ends among the thousands of data reported by the IMU and this could be a solution for future studies. Going further, recent pioneering studies have combined the corresponding gate crossing timing information obtained by a M-BTS fused with inertial sensor information and gate location details to validated a system that estimates the 3D kinematics of the skiers’ center of mass during competition events in alpine skiing [27]. It has also been shown that the M-BTS fusion with low-cost GNSS receivers can improve the accuracy of determining the kinematics of the center of mass in alpine ski racing [38].

Limitations of M-BTS: A total of 290 gate crossings were detected with the proposed system, 100% of the total number of crossings in both sport contexts. However, it should be noted that in the linear running test, five athletes had to be discarded due to poor detection of the first peak in the magnetometer signal. This occurred because the starting line of the running test coincided with the position of the first bar magnet, the fact of being located in the same place as the bar magnet caused the detection of a constant magnetic field, with no peaks appearing. An alternative should be considered if a magnet needs to be placed right at the starting location.

The results showed relatively small mean time differences in both contexts, but special attention should be paid to the linear run test, where a speed-dependent dispersion of time differences was observed, which decreased at higher speeds. Supej et al. [12] also detected this velocity dependence in a running test with the GNSS-based approach. In this sense, in competitive contexts, the use of photocells is recommended since the accuracy of the result does not depend on speed. In the case of skiing, we cannot affirm that the accuracy of the result is speed-dependent since it has not been tested in this study. It would be necessary for future studies to verify this aspect.

It should be noted that in the slalom discipline, the IMU passes very close to the magnet, as long as the skier’s technical level does not limit them to pass close to the pole. A M-BTS system has also been tested with elite athletes in giant slalom [27]. However, with lower-level athletes and especially in speed disciplines, gates may be passed with greater distances than 1 m in which magnetometer may not detect the magnetic field of the magnets. Other authors have suggested that this could be countered by increasing the strength of the magnetic field of the magnets or by placing several magnets along a line perpendicular to the intended ski trajectories [13,27]. In the context of running, as there is no need to bury the magnets for safety reasons, the magnets are placed directly on the ground surface as if they were sports cones or even, if necessary, could be lifted with sports poles to bring them closer to the athlete’s IMU. In this sense, there is no distance limitation for similar sports contexts. 

Nowadays, the use of software is necessary to treat the IMU magnetometer signal. The time between peaks is obtained instantaneously as there is a peak detection monitor. It should be noted that there would be an option to have the data loaded in real time with the help of an antenna that connects the IMU device to the computer via Bluetooth. However, the computer and the IMU cannot be far away, which in the context of skiing is unlikely. One possible solution would be to develop a system where timing data could be transmitted to a smartphone or tablet via Bluetooth. There are already companies that have implemented this type of system (Freelap USA, Pleasanton, CA, USA).

## 5. Conclusions

In technical skiing disciplines, such as slalom, as well as in sprint disciplines in athletics, the type of short-duration actions required accurate and reliable timing measurement systems. Currently, one of the most widely used timing technologies in these sport contexts is the photocell. However, this system has some limitations, such as time-consuming preparation and the high economic cost among others. In recent years, the possibility of using a M-BTS to measure short-term sports actions has become a good alternative. Therefore, this study evaluates the validity of a M-BTS for timing short-duration sport actions using an IMU’s magnetometer. The M-BTS has proven to be valid for timing sports actions ranging from 0.6 to 7.9 s in a linear sprint running test and from 0.9 to 3.2 s in an alpine ski slalom. The proposed system could be used instead of photocells during regular training in different sport contexts considering that subtle performance changes below reported 95% error-ranges would not be detected. The M-BTS can be applied in different sports contexts, regardless of where they are practiced, maintaining the technical movements of athletes. As future applications, since the M-BTS could be used to section the time series of the other integrated sensors in the wearable device, such as the accelerometer, the gyroscope, or the GNSS receiver, it could be possible to estimate other 3D kinematic parameters during the course of an event.

## Figures and Tables

**Figure 1 sensors-21-05773-f001:**
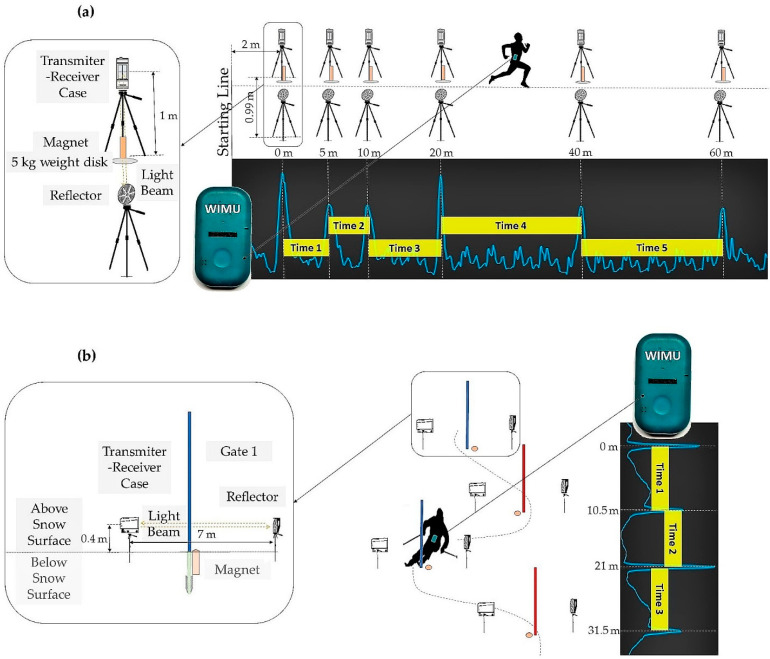
Diagram of (**a**) an athlete performing the 60-m linear sprint running test, and (**b**) a skier going down the slalom wearing an IMU device at the lower back and passing through the gates created with the photocells and magnets. The gray boxes represent the peak-shaped magnetic field recorded by the IMU’s magnetometer when passing close to the magnets at a given speed. The small yellow boxes represent the section times obtained between peaks. The photographs show the placement of the magnets in both environments.

**Figure 2 sensors-21-05773-f002:**
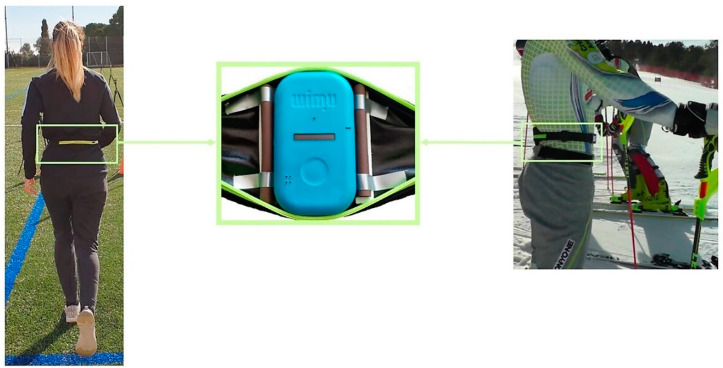
IMU device fixed on the athlete/skier’s lower back, at the L4–L5 level using an adjustable sports lycra belt.

**Figure 3 sensors-21-05773-f003:**
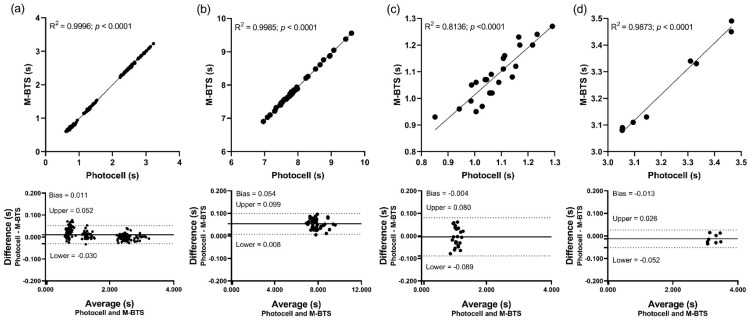
Scatterplots (above) and Bland-Altman plots (below) of time measured by M-BTS against the actual time obtained from Photocells. The diagonal line represents the identity line (slope = 1; intercept = 0). The solid line within the graph represents the bias. The broken lines represent the upper and lower 95% limits of agreement. (**a**) Belong to the linear running test and represent the analysis of the different section times, (**b**) belong to the linear running test and represent the analysis of the total time, (**c**) belong to de ski slalom and represent the analysis of the different section times (**d**) belong to de ski slalom and represent the analysis of the total time. M-BTS, magnet-based timing system.

**Table 1 sensors-21-05773-t001:** Mean difference of the time obtained through the M-BTS against the actual time obtained from Photocells.

		Distance	Speed	Photeocell Time	M-BTS Time	Mean Difference	2.5th Percentile	97.5th Percentile	95% Error-Range
		(m)	(Km/h)	(s)	(s)	(s)	(s)	(s)	(s)
Running	Section 1 (0–5 m)	5.00	22.92 ± 1.64	0.789 ± 0.058	0.757 ± 0.053	0.032 ± 0.023	−0.007	0.068	0.076
	Section 2 (5–10 m)	5.00	26.11 ± 1.72	0.693 ± 0.050	0.677 ± 0.047	0.016 ± 0.018	−0.019	0.046	0.065
	Section 3 (10–20 m)	10.00	27.88 ± 1.81	1.297 ± 0.088	1.288 ± 0.090	0.008 ± 0.015	−0.009	0.041	0.049
	Section 4 (20–40 m)	20.00	28.63 ± 2.18	2.530 ± 0.208	2.531 ± 0.207	−0.0003 ± 0.013	−0.023	0.020	0.043
	Section 5 (40–60 m)	20.00	27.84 ± 2.28	2.640 ± 0.228	2.607 ± 0.229	−0.003 ± 0.012	−0.024	0.031	0.055
	Total (0–60 m)	60.00	27.44 ± 1.96	7.913 ± 0.606	7.860 ± 0.608	0.053 ± 0.023	0.011	0.087	0.077
Skiing	Section 1 (G1–2)	10.50	33.60 ± 3.10	1.133 ± 0.103	1.144 ± 0.090	−0.011 ± 0.049	−0.065	0.061	0.126
	Section 2 (G2–3)	10.50	38.03 ± 3.03	0.999 ± 0.073	0.983 ± 0.035	0.017 ± 0.047	−0.079	0.058	0.137
	Section 3 (G3–4)	10.50	34.23 ± 1.91	1.107 ± 0.064	1.126 ± 0.057	−0.019 ± 0.027	−0.048	0.034	0.082
	Total (G1–4)	31.50	35.09 ± 1.87	3.240 ± 3.253	3.253 ± 0.169	−0.013 ± 0.020	−0.035	0.015	0.050

Values are means ± SD. M-BTS, magnet-based timing system. For the 60-m linear sprint running test: Section 1 is the time elapsed from 0 to 5 m, Section 2 is the time elapsed from 5 to 10 m, Section 3 is the elapsed time from 10 to 20 m, Section 4 is the time elapsed from 20 to 40 m, Section 5 is the time elapsed from 40 to 60 m, Total Time is the sum of the five sections. For the ski slalom: Section 1 is the time elapsed between gate 1 and 2 (G1–2), Section 2 is the time elapsed between gate 2 and 3 (G2–3), Section 3 is the elapsed time between gate 3 and 4 (G3–4), Total Time is the sum of the three sections.

## Data Availability

Dataset can be downloaded from https://www.mdpi.com/article/10.3390/s21175773/s1, Appendix A: Study database.

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
