# Peer review of "Validity of a Magnet-Based Timing System Using the Magnetometer Built into an IMU"

_sensors, 2021, doi:10.3390/s21175773_

Round 1
Reviewer 1 Report
The authors propose a novel application of a conventional and well-known IMU sensor. The work presents a suitable sample and a well-designed methodology. The statistic analysis is robust. The results are following the hypothesis. There are a few issues to address:
Major Issues:
- A description of the signal treatment is necessary. Were the peaks/time between peaks manually calculated? Is there an algorithm? Was the signal pre-processed (filtered, outlier detection)? This information is necessary to assure the reproducibility of the work.
- Conclusion session should be expanded. A direct conclusion is indeed appreciated but I suggest at least reintroduce the work problem and answer the hypothesis with details. More information such as possible future works and limitations are also welcome.
Minor Issues:
-There is room for improvement in Figure 1.
* Figure has a low graphical quality and it is hard to see the details.
* Colors in the graph aren't helping (light blue in gray might not be the best color choice for a graph).
* Pictures are a bit confusing. Maybe the big picture (literally) could be a better choice to show the details. Adding labels may also help.
* Maybe split Figures to have better details?
* In general I suggest rethinking Figure 1, as it is an important graphical illustration of the methodology.
Author Response
Response to Reviewer 1 Comments
Point 1: The authors propose a novel application of a conventional and well-known IMU sensor. The work presents a suitable sample and a well-designed methodology. The statistic analysis is robust. The results are following the hypothesis. There are a few issues to address:
Response 1: We appreciate the reviewer comments.
Point 2: A description of the signal treatment is necessary. Were the peaks/time between peaks manually calculated? Is there an algorithm? Was the signal pre-processed (filtered, outlier detection)? This information is necessary to assure the reproducibility of the work. 

Response 2: The authors agree with the reviewer. This information is relevant and was missing in the manuscript. We have added it in Lines 144-156: “The section times collected by the M-BTS were not obtained instantly, but were calculated afterwards. Each time an athlete or skier passed near a bar magnet, a peak appeared in the magnetometer time series. SPRO software (Realtrack Systems, Almeria, Spain) was used to download the data stored in the WIMU in order to calculate the elapsed time between peaks (Figure 1). The magnetometer source signal was used as the dominant peaks corresponded to the gate locations. Since the magnets were placed vertically, the magnetic field came from the ground upwards. For this magnet placement, the vertical x-axis of the WIMU magnetometer was the one that collected the highest magnitude peaks and was therefore selected. The signal was loaded into the program workspace and the selection tool was used to select the signal chunks corresponding to the sprints and the downhill ski slaloms to be analysed. The SPRO peak detection tool was then manually configured to detect the maximum positive peaks lasting between 200 and 2000 ms and the time between peaks was automatically detected.”
To be sure of the correct calculation of the peak detection, the Excel report generated by SPRO was uploaded to a routine programmed in Matlab (The MathWorks, Massachusetts, USA) with the aim to detect the time elapsed between peaks. The match turned out to be identical to the hundredth of a second.
Point 3: Conclusion session should be expanded. A direct conclusion is indeed appreciated but suggest at least reintroduce the work problem and answer the hypothesis with details. More information such as possible future works and limitations are also welcome
Response 3: The authors agree with the reviewer and they have made the relevant modifications in Lines 405-422: “In technical skiing disciplines, such as slalom, as well as in sprint disciplines in athletics, the type of short-duration actions required accurate and reliable timing measurement systems. Currently, one of the most widely used timing technologies in these sport contexts is the photocell. However, this system has some limitations as time consuming preparation and the high economic cost among others. In recent years, the possibility of using a M-BTS to measure short-term sports actions has become a good alternative. Therefore, this study evaluates the validity of a M-BTS for timing short-duration sport actions using a WIMU’s magnetometer. The M-BTS has proven to be valid for timing sports actions ranging from 0.6 to 7.9 s in a linear sprint running test and from 0.9 to 3.2 s in an alpine ski slalom. The proposed system, could be used instead of photocells during regular trainings in different sport contexts considering that subtle performance changes below reported 95% error-ranges would not be detected. The M-BTS can be applied in different sports contexts, regardless of the place where it is practiced, maintaining the technical movements of athletes.
As future applications, since the M-BTS could be used to section the time series of the other integrated sensors in the wearable device, such as the accelerometer, the gyroscope or the GNSS receiver, it could be possible to estimate other 3D kinematic parameters during the course of an event.”
Point 4: There is room for improvement in Figure 1.
* Figure has a low graphical quality and it is hard to see the details.
* Colors in the graph aren't helping (light blue in gray might not be the best color choice for a graph).
* Pictures are a bit confusing. Maybe the big picture (literally) could be a better choice to show the details. Adding labels may also help.
* Maybe split Figures to have better details?
* In general I suggest rethinking Figure 1, as it is an important graphical illustration of the methodology.
Response 4: Reviewed and amended. Figure 1 has been improved. As the reviewer suggested, it was an important graphical illustration of the methodology and the details were poorly visible.
- The authors have increased its quality and made all its images larger.
- The authors have modified the colours to help the reader to better discern all the details, especially the colours of the magnetometer signal that were not visible in the previous image.
- The authors have changed the two photos of the placement of the magnets for graphic drawings where the details can be better appreciated. In addition, as suggested by the reviewer, we have added labels for better understanding.
- It has been decided to keep only one figure, the authors believe that with all the changes made it has not been necessary to divide it in two.
In Line 113, there is the new Figure 1 Updated. The authors hope that now all the details of this figure will be well appreciated.
In addition, although not suggested by the reviewer, the quality of Figure 2 has also been improved. Line 159.
Finally, the authors would like to thank all the reviewer's contributions that have helped to improve this article.
Reviewer 2 Report
The authors are commended on conducting an interesting study to determine a new technique for quantifying time during an event. This certainly is an emerging area. However, as a reader, I was expecting the technique to lead to a continuous measure of a time-based parameter such as velocity. The introduction discussed the limitation of time only being recorded at specific intervals and it does not seem that the magnetic based system improves on that. However, since the main device worn by the subject is an IMU, it would seem possible to use the IMU data to determine velocity between gates. That being said, it is understood now that that was not the purpose of the study. It is suggested, however, to discuss how this technique might lead to an innovative way to measure small intervals of time during the course of an event.
The one methodological issue that was not clear was the detection of peaks. It is understood through Figure 1a that there were dominant peaks that corresponded to the gate locations. However, there were many more local peaks between gates. The reader would benefit from a description of the processing algorithm that allowed only the peaks that corresponded to the gates to be detected.
Author Response
Response to Reviewer 2 Comments
Point 1: The authors are commended on conducting an interesting study to determine a new technique for quantifying time during an event. This certainly is an emerging area. However, as a reader, I was expecting the technique to lead to a continuous measure of a time-based parameter such as velocity. The introduction discussed the limitation of time only being recorded at specific intervals and it does not seem that the magnetic based system improves on that. However, since the main device worn by the subject is an IMU, it would seem possible to use the IMU data to determine velocity between gates. That being said, it is understood now that that was not the purpose of the study. It is suggested, however, to discuss how this technique might lead to an innovative way to measure small intervals of time during the course of an event.
Response 1: The authors are grateful for the assessment made by this reviewer. He/She is absolutely correct that, despite having commented in the introduction on the limitation that time is only recorded at specific intervals, the proposed system does not differ from photocells in this regard. However, The M-BTS can be used to cut the time series of all the other integrated sensors in the WIMU such as the accelerometer, gyroscope or GNSS receiver. Unlike photocells, M-BTS is a new validated application on the use of these wearable sensors, so they are not two separate systems, but M-BTS and WIMU are part of the same system.
It has also been suggested to discuss how the use of the M-BTS system could lead to an innovative way of measuring small time intervals during the course of an event, since the wearable device worn by the athletes is an IMU. As there are recent pioneering studies contributing knowledge on this line of research, the authors have thought it convenient to include them in the discussion. Lines 355-361: “Going further, recent pioneering studies have combined the corresponding gate crossing timing information obtained by a M-BTS fused with inertial sensor information and gate location details to validated a system that estimates the 3D kinematics of the skiers center of mass during competition events in alpine skiing [27]. It has also been shown that the M-BTS fusion with low-cost GNSS receivers can improve the accuracy of determining the kinematics of the center of mass in alpine ski racing [38].” The reviewer's contribution has also been noted in the conclusions section as a future research line. Lines 419-422: “As future applications, since the M-BTS could be used to section the time series of the other integrated sensors in the wearable device, such as the accelerometer, the gyroscope or the GNSS receiver, it could be possible to estimate other 3D kinematic parameters during the course of an event.”
Point 2: The one methodological issue that was not clear was the detection of peaks. It is understood through Figure 1a that there were dominant peaks that corresponded to the gate locations. However, there were many more local peaks between gates. The reader would benefit from a description of the processing algorithm that allowed only the peaks that corresponded to the gates to be detected.
Response 2: The authors agree with the reviewer. This information is relevant and was missing in the manuscript. We have added it in Lines 144-156: “The section times collected by the M-BTS were not obtained instantly, but were calculated afterwards. Each time an athlete or skier passed near a bar magnet, a peak appeared in the magnetometer time series. SPRO software (Realtrack Systems, Almeria, Spain) was used to download the data stored in the WIMU in order to calculate the elapsed time between peaks (Figure 1). The magnetometer source signal was used as the dominant peaks corresponded to the gate locations. Since the magnets were placed vertically, the magnetic field came from the ground upwards. For this magnet placement, the vertical x-axis of the WIMU magnetometer was the one that collected the highest magnitude peaks and was therefore selected. The signal was loaded into the program workspace and the selection tool was used to select the signal chunks corresponding to the sprints and the downhill ski slaloms to be analysed. The SPRO peak detection tool was then manually configured to detect the maximum positive peaks lasting between 200 and 2000 ms and the time between peaks was automatically detected.”
To be sure of the correct calculation of the peak detection, the Excel report generated by SPRO was uploaded to a routine programmed in Matlab (The MathWorks, Massachusetts, USA) with the aim to detect the time elapsed between peaks. The match turned out to be identical to the hundredth of a second.
Finally, the authors hope to have been able to respond appropriately to all the revisions and would like to thank the reviewer for having contributed to the improvement of the article.